# Optimising a method for aragonite precipitation in simulated biogenic calcification media

**Celeste Kellock** [1¤]*, **Maria Cristina Castillo Alvarez**[1], **Adrian Finch**[1], **Kirsty Penkman**[2], **Roland Kröger**[3], **Matthieu Clog** [4], **Nicola Allison**[1]

**1** School of Earth and Environmental Sciences, University of St. Andrews, St. Andrews, United Kingdom,
**2** Department of Chemistry, University of York, York, United Kingdom, **3** Department of Physics, University of York, York, United Kingdom, **4** Scottish Universities Environmental Research Centre, Glasgow, United Kingdom

¤ Current address: Department of Biological and Environmental Sciences, University of Stirling, Stirling, Scotland, United Kingdom
* celeste.kellock@stir.ac.uk

**Data Availability Statement:** All relevant data are within the paper and its Supporting Information files.

**Funding:** This work was supported by the Leverhulme Trust (Research project grant 2015-

## Abstract

Resolving how factors such as temperature, pH, biomolecules and mineral growth rate influence the geochemistry and structure of biogenic $CaCO_3$, is essential to the effective development of palaeoproxies. Here we optimise a method to precipitate the $CaCO_3$ polymorph aragonite from seawater, under tightly controlled conditions that simulate the saturation state ($\Omega$) of coral calcification fluids. We then use the method to explore the influence of aspartic acid (one of the most abundant amino acids in coral skeletons) on aragonite structure and morphology. Using $\geq$200 mg of aragonite seed (surface area 0.84 m$^2$), to provide a surface for mineral growth, in a 330 mL seawater volume, generates reproducible estimates of precipitation rate over $\Omega_{\text{aragonite}}$ = 6.9–19.2. However, unseeded precipitations are highly variable in duration and do not provide consistent estimates of precipitation rate. Low concentrations of aspartic acid (1–10 µM) promote aragonite formation, but high concentrations ($\geq$ 1 mM) inhibit precipitation. The Raman spectra of aragonite precipitated *in vitro* can be separated from the signature of the starting seed by ensuring that at least 60% of the analysed aragonite is precipitated *in vitro* (equivalent to using a seed of 200 mg and precipitating 300 mg aragonite *in vitro*). Aspartic acid concentrations $\geq$ 1mM caused a significant increase in the full width half maxima of the Raman aragonite $v_1$ peak, reflective of increased rotational disorder in the aragonite structure. Changes in the organic content of coral skeletons can drive variations in the FWHM of the Raman aragonite $v_1$ peak, and if not accounted for, may confuse the interpretation of calcification fluid saturation state from this parameter.

## 1. Introduction

Marine biogenic carbonates are invaluable archives of past climate information, potentially recording information on seawater composition, temperature and pH in their geochemistry and structure (e.g. the Sr/Ca ratio of aragonite coral skeletons and the Mg/Ca ratio of calcite

268 to NA, RK, and KP) and the UK Natural Environment Research Council (NE/S001417/1) to NA, KP, RK, MC and AF. The Raman Microscope is supported by the EPSRC (Light Element Analysis Facility Grant EP/T019298/1 and Strategic Equipment Resource Grant EP/R023751/1). The funders had no role in study design, data collection and analysis, decision to publish, or preparation of the manuscript.

**Competing interests:** The authors have declared that no competing interests exist.

foraminifera tests are influenced by seawater temperatures) [1, 2]. Rotational disorder in the $CaCO_3$ structure is inferred to reflect changes in the dissolved inorganic carbon (DIC) chemistry of the media used for biomineralisation in biogenic carbonates, and may reflect changes in ocean carbonate saturation state [3, 4]. Carbonate proxies are instrumental in shaping our understanding of past climate and critical in validating global climate models for predicting 21st century climate change [5]. In spite of this potential, the relationships between environment, $CaCO_3$ geochemistry and structure are poorly understood. $CaCO_3$ geochemistry and structural order can be affected by temperature [6, 7], pH and [DIC] [8–10], the presence of organic ligands [11] (which occur in biogenic carbonates) and $CaCO_3$ precipitation rate [12, 13]. Resolving how these multiple factors influence $CaCO_3$ geochemistry and structure, separately and in combination, is essential to the effective development of palaeoproxies.

The influence of temperature and solution chemistry on $CaCO_3$ geochemistry and structure can be explored in $CaCO_3$ precipitations *in vitro*. A range of methods have been adopted for this, including $Na_2CO_3$ addition [6, 9], $CO_2$ degassing from high $pCO_2$ (1 atm) solutions, pH-stat [9] and ammonium carbonate decomposition [12, 14]. These studies yield many useful insights into the controls on $CaCO_3$ chemistry. However, designing experiments which permit $CaCO_3$ precipitation under steady state solution conditions that are comparable to those of calcareous organism calcification sites (where the $CaCO_3$ mineral is formed) is challenging. A pH-stat can be used to maintain constant solution pH, but significant invasion (or outgassing) of $CO_2$ occurs if the solution and surrounding atmosphere are not at equilibrium and this alters the solution DIC chemistry (Fig 1. For example, [DIC] more than doubled when we

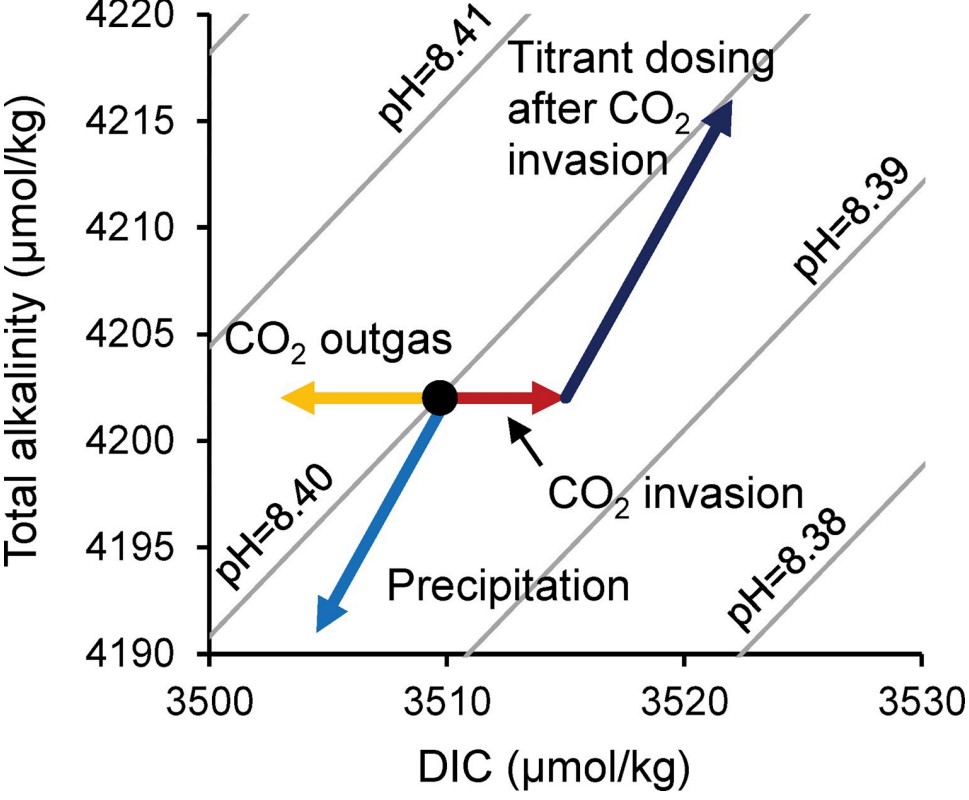

**Fig 1. Effect of processes on the DIC chemistry of seawater.** Precipitation reduces DIC, alkalinity and hence pH. Titrant dosing after precipitation returns the DIC to the starting composition (black dot). $CO_2$ invasion and outgas modify the DIC but not the alkalinity of the seawater. $CO_2$ invasion reduces pH and triggers the dosing of titrants which return the seawater to the starting pH but at a different DIC and alkalinity composition. Contours join points of equal pH (NBS scale).

precipitated aragonite by maintaining a seawater solution at pH 8.7 in an open laboratory. In some reported studies the $Ca^{2+}$ consumed during $CaCO_3$ precipitation was not replaced and solution $[Ca^{2+}]$ decreased by >70% [9] during experiments. These variations in solution pH, [DIC] and $[Ca^{2+}]$ generate large changes in solution $CaCO_3$ saturation state ($\Omega$) during the lifetime of the experiment, which in turn affect $CaCO_3$ growth rates [15] and may influence geochemistry and structure.

Here we report a study to optimise a method to precipitate aragonite under tightly controlled temperature, pH, $\Omega$, [DIC] and $[Ca^{2+}]$ conditions, similar to those of coral calcification media. We conducted precipitations both without and with a seed to provide a nucleation surface for the growth of aragonite *in vitro* and to permit the normalisation of precipitation rate to seed surface area [15]. We investigated the optimal seed surface area in our apparatus to provide consistent results and tested the dynamics of precipitations in both artificial and natural seawaters. We analysed precipitated aragonites using Raman spectroscopy to test how variations in the proportion of seed versus aragonite precipitated *in vitro* affect structure. Raman spectroscopy measures the change in photon frequency (Raman shift) as monochromatic light is scattered inelastically by interaction with molecular vibrations within a material. It is used to identify $CaCO_3$ polymorph [16], detect disorder in the $CaCO_3$ lattice [3, 4, 16], explore the distribution of organic materials [16] and infer Mg content in biogenic carbonates [3, 4]. Finally, we demonstrate use of our optimised methodology to test the effect of varying the concentration of aspartic acid on aragonite precipitation rate and structure. Aspartic acid is the most common amino acid in Scleractinia coral skeletons [17, 18] as a key component of biomineralisation proteins [19]. Resolving the role that biomolecules play in aragonite precipitation and structure is critical to a full understanding of the biomineralisation process and how this may change under different climate conditions.

## 2. Methods

### 2.1 Precipitation apparatus

To address the aims of the present study, we created an automated precipitation apparatus in which the chemical parameters are maintained at constant values throughout the experiment. We precipitated aragonite from seawater solutions with pH and [DIC] altered from typical seawater values. Precipitation of $CaCO_3$ consumes DIC and $Ca^{2+}$ and reduces solution pH (Fig 1). The pH of the solution was constantly monitored using a high precision pH/temperature sensor (Metrohm Aquatrode Pt1000) and a pH decrease triggered an adapted Metrohm Titrando 902 titrator to add equal volumes of 0.6 M $Na_2CO_3$ and $CaCl_2$ titrants to replace the ions precipitated from solution. The $CaCl_2$ titrant was prepared as 0.594 M $CaCl_2$ + 0.006 M $SrCl_2$ to ensure replacement of both the Ca and the Sr which can substitute for Ca in the aragonite lattice [20].

For each experiment 330–340 mL of filtered seawater (0.2 μm polyether sulfone filter) was measured into a high-density polyethylene (HDPE) plastic beaker (total volume ~360 mL) and capped with an ethylene tetrafluoroethylene lid with multiple ports. A pH sensor, a propeller stirrer, a gas tube and the 2 titrant dosing tubes were inserted through the lid and into the headspace (gas tube) or seawater (all others). The HDPE beaker was maintained at 25°C by immersion in a heated circulating water bath (Optima TC120) fitted with a cooling coil. To avoid invasion or outgassing of $CO_2$ which alter seawater pH (Fig 1) and DIC speciation, the precipitating solution was maintained under a headspace with a filtered ambient air gas stream ($pCO_2$ = ~416 μatm) and the pH and DIC of the seawater were adjusted to be in equilibrium with this atmosphere. At the start of each experiment seawater [DIC] was increased with the addition of 0.6 M $Na_2CO_3$ and the pH adjusted to the required value by addition of 1.0 M HCl

or NaOH. Once pH was stable the dry, weighed aragonite seed (if used) was suspended in 1 mL of the seawater solution before being added to the reaction vessel. The titration usually proceeded until 5 mL of each titrant were added to the reaction vessel resulting in the precipitation of ~300 mg of aragonite. The pH sensor was calibrated each week with fresh buffers. Over the course of the week the pH of the buffers changed by <0.003 pH units. At the end of each experiment the sensor, beaker and propeller stirrer were submerged in 0.1 M HCl to dissolve any precipitate and then rinsed thoroughly with deionized water. The sensor was acclimated in seawater for at least 30 minutes before reuse.

## 2.2 Testing the effects of $\Omega$ and seed mass on aragonite precipitation

Experiments were conducted over a range of $\Omega$ and in either artificial or natural seawater (Table 1). Natural seawater was collected from the shore in Crail, Fife, UK, filtered and stored in a blacked-out 1000 L HDPE container for several weeks before use. Artificial seawater was made according to reference [21]. Both waters were bubbled with atmospheric air sourced from outside the building ([$CO_2$] ≈ 410–420 ppm) before use. The total alkalinity and DIC (after bubbling) were determined by automated Gran titration (Metrohm, 888 Titrando) and using an Apollo SciTech (AS-C3) DIC Analyser [22]. Both instruments were calibrated with CRM (certified reference material) (A. Dickson, Scripps Institution of Oceanography) and yielded precision (standard deviation) of multiple analyses of <0.2% in each case. Salinity was estimated from conductivity measured with a Thermo Orion 5 star pH/RDO/conductivity meter calibrated with NIST standards.

Precipitations were either conducted using no seed or using 50, 100, 200 or 400 mg of an aragonite seed produced by wet grinding pieces of a *Porites lutea* coral skeleton in an agate ball mill. The seed had a surface area of 4.2 $m^2$ $g^{-1}$ determined by the Brunauer-Emmett-Teller technique, assuming a density of aragonite of 2.94 g $cm^{-3}$. $\Omega_{aragonite}$ (hereafter abbreviated to $\Omega$) were calculated using [$Ca^{2+}$] (Table 1) and [$CO_3^{2-}$] and the solubility product ($K_{sp}$) of aragonite at 25˚C and 1 atmosphere [23]. Seawater [$CO_3^{2-}$] is calculated using CO2 sys v2.1 [reference 24] from seawater [DIC] and $pH_{NBS}$ using the equilibrium constants for carbonic acid and $KHSO_4$ [references 25, 26] and total [B] [reference 27]. The small variation in [$Ca^{2+}$] between the artificial and natural seawater altered $\Omega$ by ≤0.2 and we consider this insignificant in terms of the $\Omega$ range studied. Experiments were conducted at a $pH_{NBS}$ = 8.337 with DIC = 3000 µmol $kg^{-1}$, $pH_{NBS}$ = 8.445 with DIC = 4000 µmol $kg^{-1}$ and $pH_{NBS}$ = 8.564 with DIC = 5500 µmol $kg^{-1}$. Mean $\Omega$ of the natural and artificial seawaters at each pH were 6.9, 11.3 and 19.2 respectively. $\Omega$ of coral calcification media is estimated to be ~12 based on microsensor measurements of media pH and [$CO_3^{2-}$] [reference 28].

Using this methodology, pH variations within a precipitation were <0.002 pH units (1σ). Seawater temperatures within and between precipitations varied by <0.3˚C. To confirm the seawater conditions, [DIC] was measured at varying time points in a subset of precipitations after filtering the seawater through a 0.2 µm polyethersulfone filter.

**Table 1. Chemistry of the waters used for precipitations.** [$Ca^{2+}$] and [$Mg^{2+}$] is estimated for artificial seawater (based upon composition) and measured (by ICP-OES) for natural seawater.

|  | Natural seawater | Artificial seawater |
|---|---|---|
| Total alkalinity (µmol $kg^{-1}$) | 2208 | 2309 |
| DIC (µmol/$kg^{-1}$) when bubbled at ambient p$CO_2$ | 1933 | 2015 |
| Salinity | 34.1 | 33.7 |
| [$Ca^{2+}$] mM | 10.1 | 9.9 |
| [$Mg^{2+}$] mM | 51 | 53 |

The experiment durations were long for unseeded experiments (up to 32 hours) and these were only conducted at $\Omega$ = 6.9 and 19.9 in artificial seawater. Durations were also long for experiments using 50 mg of seed at $\Omega$ = 6.9 (up to 12 hours) and these titrations were stopped after dosing of 2 mL of each titrant to prevent $CO_2$ invasion. Other seeded experiments took from 13 minutes to ~8 hours.

### 2.3 Testing the effect of seed on Raman spectroscopy signature of *in vitro* precipitate

We conducted multiple precipitation experiments at $\Omega$ = 11.3 ($pH_{NBS}$ = 8.445, DIC = 4000 µmol $kg^{-1}$) in artificial seawater using 200 mg of the aragonite seed and varying the volume of each titrant dosed into the solution from 1.67 mL to 13.3 mL. Changing the titrant volume resulted in the amount of aragonite precipitated *in vitro* varying between 100 and 800 mg. At the end of each experiment, the $CaCO_3$ in the reaction vessel (a mixture of the original seed and the experimental precipitate) was collected by filtration onto a 0.2 µm polycarbonate track etched membrane filters (Cytiva Whatman), rinsed with deionised water and dried at 40˚C. Raman spectra of the precipitates and the original seed were collected between 100–1311 wave numbers with a Renishaw In-Via Qontor Raman Microscope using a NIR 300 mW 785 nm solid state laser with a 1200 $cm^{-1}$ grating. We use Raman data to confirm the $CaCO_3$ polymorph formed and to explore the degree of internal rotational disorder. Here we focus on the $v_1$ peak, typically the highest intensity peak in the aragonite spectrum. The laser was focused to ~10 x 1 µm and positioned lengthwise along the edges of particles i.e. where *in vitro* precipitation occurs. The instrument was calibrated by measuring an internal Si standard. Multiple spectra (n = 10 or 11) were collected from each sample using a 5% laser power and summing 10 scans with a total acquisition time of 20 s. The spectra were processed using OriginLabs software and the full width half maxima (FWHM) of the $v_1$ peak were estimated following a Voigt fit which typically yields a better coefficient of determinations than a Gaussian fit and provides a better model of individual vibration bands combined with instrumental artefacts [29]. We tested for variations in the peak position and FWHM between samples using one way ANOVA followed by Tukey's pairwise comparison.

### 2.4 Testing the effect of aspartic acid on precipitation rate and Raman signature

To test our optimised method we conducted aragonite precipitations in artificial seawater at [aspartic acid] from 1 µM to 8.9 mM. All experiments were conducted at $pH_{NBS}$ = 8.445, [DIC] = 4000 µmol kg $^{-1}$. $\Omega$ = 11.2 and using a seed mass of 200 mg. The aspartic acid (if used) was added to the precipitation seawater before altering the DIC by either suspending the mass of amino acid (L-aspartic acid, Sigma Life Science, >98% purity) required to obtain the final seawater concentration in 1 mL of seawater and then pipetting this into the reaction vessel (for 1 and 8.7 mM) or by dissolving a known mass of amino acid in seawater to produce a stock solution and pipetting aliquots of this into the reaction vessel (for ≤100 µM). Stock aspartic acid solutions were used within 2 hours and then discarded. Precipitates were collected by filtration, dried and characterised by Raman spectroscopy as before. We tested for variations in the peak centre and FWHM of the aragonites using one way ANOVA followed by Tukey's pairwise comparison.

### 2.5 Scanning electron microscopy

Precipitates were examined by scanning electron microscopy using a JEOL 7800F using a 3kV accelerating voltage at the University of York. Samples were mounted on carbon tabs and no coating was applied.

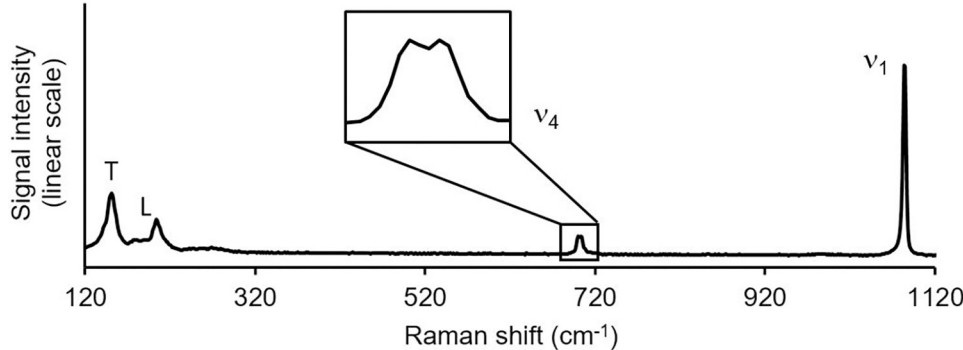

**Fig 2. Example Raman spectrum.** The material is identified as $CaCO_3$ based on the strong $\nu_1$ peak at ~1084 cm⁻¹ and as aragonite based on the dual peak ($\nu_4$) between 700–710 cm⁻¹ [reference 30].

## 3. Results and discussion

### 3.1 Precipitate characterisation

Raman spectroscopy (Fig 2) (conducted on at least one precipitate produced under each set of conditions) confirmed that the original seed and precipitates were aragonite. Aragonite has lower symmetry than calcite and hence a single peak in rhombohedral calcite appears as characteristic doublets in orthorhombic aragonite between 700–710 cm-1 [30]. The Raman spectrum of solid aspartic acid indicates that the amino acid has multiple peaks between 120–1120 cm⁻¹ including one that coincides with the aragonite ν1 peak (S1 Fig). However, the large aspartic acid peak at ~936 cm⁻¹ was never observed in the aragonite spectra and we conclude that any contribution of aspartic acid to the aragonite ν1 peak is insignificant.

### 3.2 Precipitation in unseeded experiments

We observed precipitation of aragonite in unseeded experiments in artificial seawater at both $\Omega = 6.9$ and $19.2$ (Fig 3). In some of these experiments the titration profile follows a smooth curve and the dosing rate accelerated as the precipitation proceeded (e.g. T1 Fig 3A and 3C). In other experiments (e.g. Fig 3A and 3C, T2) there was a period of relatively little dosing (a lag period) followed by increasingly rapid titrant addition. In one experiment, this lag period exceeds 28 hours (Fig 3). We measured the [DIC] at the start and at a midpoint in the lag of 2 of the titrations (Fig 3C) and observed an increase in DIC between these points. To identify the source of this DIC increase we used $CO_2$ sys to estimate the total alkalinity of the seawater at the start and the lag midway point from the measured solution $pH_{NBS}$ and [DIC]. The total alkalinity increased by 1186 and 480 μmol kg⁻¹ at $\Omega = 6.9$ and 19.2 respectively. At these midway points the titrator had dosed 0.318 and 0.176 mL of 0.6 M $Na_2CO_3$ respectively, sufficient to increase the total alkalinity of a 330 mL volume by 1156 and 640 μmol kg⁻¹. The good agreement between the observed alkalinity increases and that predicted to occur due to the titrant dosing suggests that little $CaCO_3$ precipitation occurs during the lag period. [DIC] increased by 1042 and 395 μmol kg⁻¹ at the midway lag point in $\Omega = 6.9$ and 19.2 respectively, although titrant dosing is predicted to increase [DIC] by 578 and 320 μmol kg⁻¹. The observed increase in [DIC] above that predicted from titrant dosing indicates an invasion of atmospheric $CO_2$ from the atmosphere into the seawater solution, i.e. due to a minor disequilibrium. This invasion reduces the solution pH and leads to the dosing of titrant, even in the absence of aragonite precipitation. The increase in solution [DIC] due to $CO_2$ invasion is 26 and 31 μmol kg⁻¹ h⁻¹ at $\Omega = 6.9$ and 19.2 respectively.

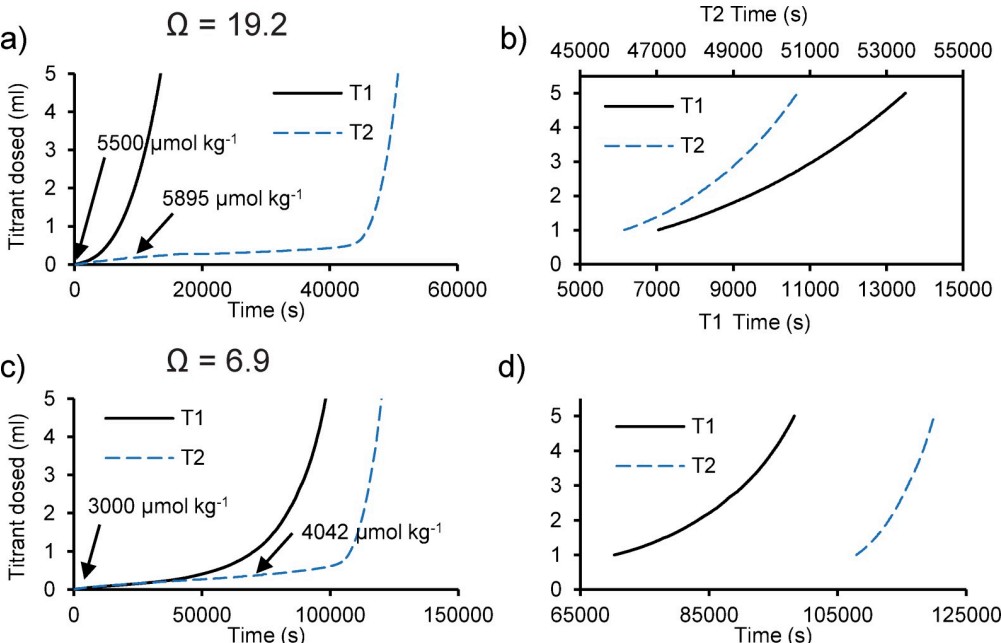

**Fig 3. Profiles of titrant volume dosed over time in duplicate unseeded experiments in artificial seawater.** a), b) $\Omega$ = 19.2 and c), d) $\Omega$ = 6.9. b) and d) showed expanded x axes to compare rates of dosing of 1–5 mLs of titrant between duplicates (T1 and T2, show data from two titrators). Measured [DIC] at the start and during 2 of the precipitations are overlaid onto the graphs.

Precipitation in the absence of an existing solid surface is termed homogenous nucleation and proceeds as constituent ions of the solid combine to form pre-nucleation clusters which aggregate and dehydrate to form amorphous nanoparticles that ultimately transform into crystals [31]. Homogeneous nucleation of $CaCO_3$ is not observed in seawater at 25˚C and below $\Omega_{\text{aragonite}} \approx 12$ (i.e. below $\Omega_{\text{calcite}} = 18$, [references 32, 33]). As minor $CO_2$ invasion into the seawater reduces solution pH and triggers titrant dosing, nucleation may occur in the vicinity of the titrant dosing tube if a higher $\Omega$ is achieved at this location. Once $CaCO_3$ has formed, this acts as a nucleation surface for subsequent growth. We observe large variations in titrant dosing rate between duplicate experiments, suggesting that the size of the precipitating $CaCO_3$ surface is inconsistent between experiments. This hampers the accurate calculation of a precipitation rate normalised to surface area which is required to investigate the influence of crystal growth rate on geochemistry and structure.

### 3.3 Precipitation in seeded experiments

**3.3.1. Effect of seed mass.** Aragonite was precipitated in all seeded experiments (Fig 4). In some precipitations the rate of titrant dosing was approximately constant resulting in a linear relationship between time and the volume of titrant dosed, while in others the rate of titrant addition increased during the experiment resulting in a curved profile. To explore the origin of these profiles we calculated the profile expected for a series of seeded precipitations assuming that precipitation occurs by epitaxial growth creating a layer all over the starting seed. We selected a precipitation rate of 617 $\mu$mol m$^{-2}$ h$^{-1}$ (as for $\Omega$ = 6.9 in natural seawater [18]), a seed surface area of 4.2 m$^2$ g$^{-1}$, a total dosing of 5 mLs of 0.6 M titrants and masses of starting seed of 400, 200, 100 and 50 mg. We worked with the following approximations: (i) the starting seed is cubic with equal dimensions of 0.486 $\mu$m (equivalent to a surface area of 4.2

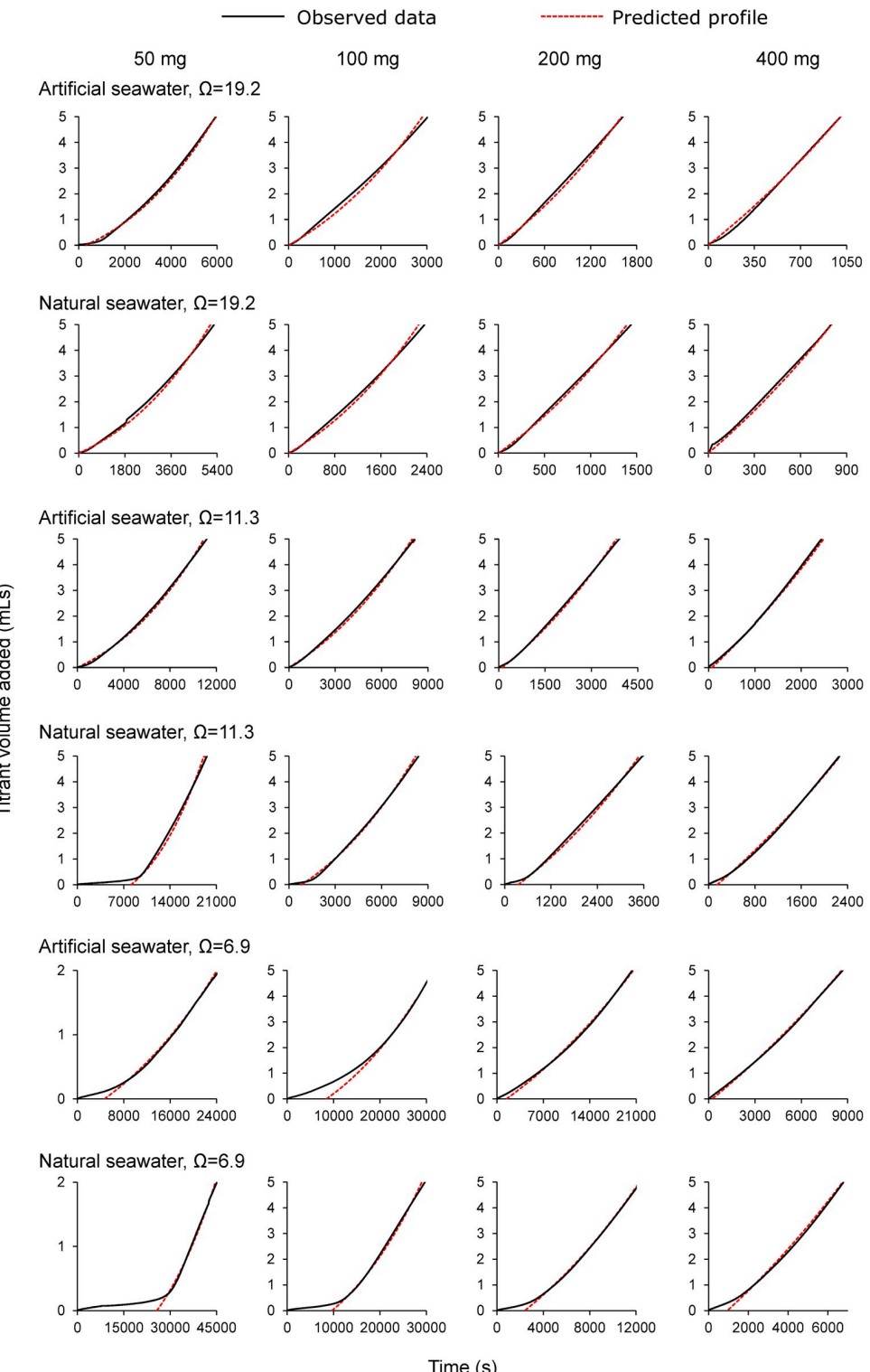

**Fig 4. Titration profiles in seeded experiments with no added aspartic acid.** Black lines show observed profile and red dotted lines show predicted profile as explained in text. N.B. the x- and y-axes are not to the same scale for all the graphs.

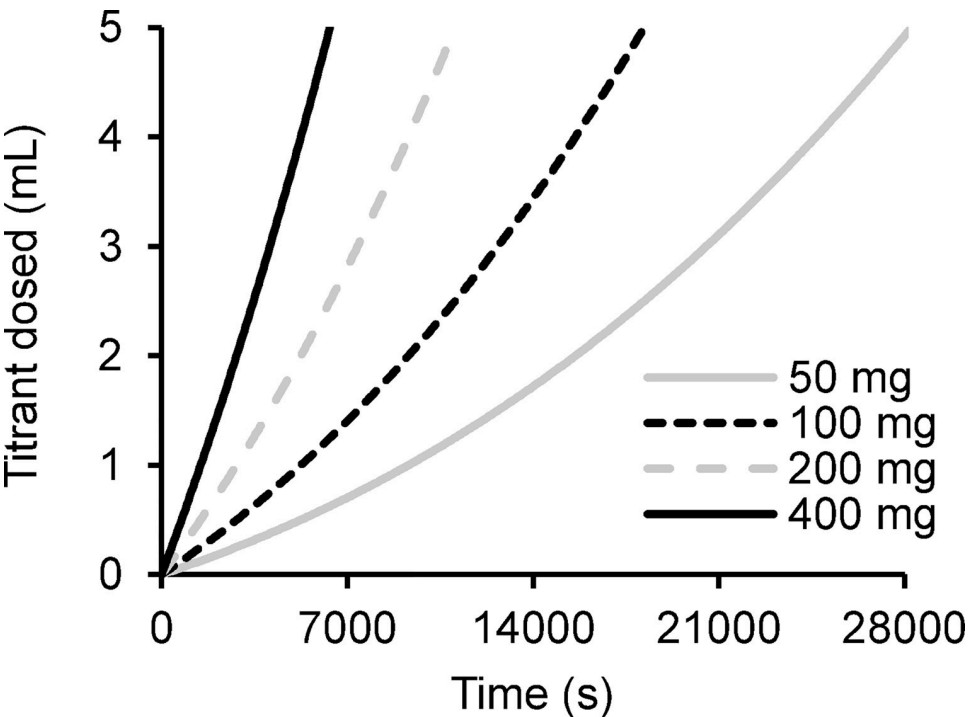

**Fig 5. Predicted titrant dosing profiles assuming that all dosing replaces ions consumed in the epitaxial growth of $CaCO_3$ over the seed.** In this example we assume a precipitation rate of 617 $\mu$mol m$^{-2}$ h$^{-1}$ onto a seed of surface area of 4.2 m$^2$ g$^{-1}$ using different masses of seed and 0.6 M titrants.

m$^2$ g$^{-1}$ assuming an aragonite density of 2.94 g cm$^{-3}$) and (ii) precipitation occurs by epitaxial growth all over the cubes. We estimate the total surface area of the seed at the start of the experiment, calculate the precipitation that will occur over a set period (typically 10–60 s depending on the amount of seed used) and estimate the volume of titrant required to replace the ions consumed in precipitation. We recalculate the surface area of the precipitate at the end of this time period and continue with these estimates until the end of the precipitation (the time point at which 5 mL of 0.6 M titrant has dosed). While it is unlikely that precipitation occurs in such an idealised manner, our approach explores if the surface area available for precipitation during the titration increases by a similar magnitude to that predicted from epitaxial growth. At higher seed mass the predicted titration profile is approximately linear (Fig 5) as precipitation has little effect on the total surface area available for aragonite growth and titrant dosing rate remains approximately constant. At low seed mass, precipitation significantly increases the surface area available for precipitation as the experiment proceeds and the rate of titrant addition accelerates during the titration making the profile more curved (Fig 5).

Some profiles exhibited lag periods; these were more frequent during precipitations that ran at lower saturation states (Fig 4). To estimate lag periods and precipitation rates from the profiles we calculated predicted titration profiles for (as for Fig 5, using the known amount of added seed), we plotted a third order polynomial regression through the prediction profile and varied precipitation rate in the prediction to achieve the highest coefficient of determination ($r^2$) between the predicted data and that observed in the precipitations [34]. For the precipitations in which a lag is observed we added a lag period to the prediction profile model. We identify the time at which 1 mL of each titrant is dosed on the experimental profiles. We subtract the time to dose 1 mL on the predicted profile from the experimental profile to determine

the lag and we add this lag period to each time point on the predicted profile. In essence we are shifting the entire predicted profile to the right so that the points where 1 mL of titrant are dosed in the experimental and predicted profiles coincide. To estimate precipitation rate we again varied precipitation rate in the prediction profile to achieve the highest coefficient of determination ($r^2$) between the observed and predicted data for the addition of 1–5 mL titrant. We overlay all predictions (with or without lag) onto Fig 5 (as red dotted lines) and, in the case of a lag, extend the profile to the y axis. We observe an excellent fit ($r^2 = 0.999$) between the predicted and observed profiles in all precipitations with no lag (all precipitations at $\Omega = 19.2$ and for those in artificial seawater at $\Omega = 11.4$). The curve in the profile at low seed mass is reproduced in the predicted profiles suggesting that the curve reflects an increase in the surface area available for $CaCO_3$ precipitation. Precipitation rates and lag periods for each set of conditions are summarised in Fig 6A and 6B respectively. Lag periods less than ~300 s could not be confidently identified as the addition of seed at the start of each experiment took 60–120 s and minor lags in the onset of dosing could reflect this delay.

Heterogeneous nucleation occurs in the presence of existing nucleation surfaces (e.g. a mineral seed), and involves the formation of a crystalline nucleus on the existing mineral surface and subsequent growth from that point. Lag periods reflect delays in this nucleation step and were most apparent at low $\Omega$ and/or seed mass and in the natural seawater (Fig 6). Lag periods >1800 s (20 minutes) occurred in both natural and artificial seawater at $\Omega = 6.9$ with 50 and 100 mg seed and in natural seawater at $\Omega = 11.4$ with 50 mg seed. Lag periods in the seeded experiments at $\Omega = 19.2$ were insignificant and at $\Omega = 6.9$ were much shorter than in unseeded experiments e.g. ~4000 s in artificial seawater with 50 mg seed compared to up to >100000 s in the unseeded analogue. This indicates that the seed provides an important surface for heterogeneous nucleation, even in the experiments with a lag period. Heterogeneous nucleation may occur on both the seed and the surface of the apparatus (beaker, sensor, titrant dosing tubes, stirrer). However the surface area of the seed (0.210 $m^2$ for a 50 mg aliquot of seed) far exceeds that of the apparatus in contact with the seawater (estimated to be <0.03 $m^2$, assuming the 73 mm diameter beaker is filled to a depth of 8.2 cm, the 12 mm diameter pH sensor is immersed to a depth of 6 cm, the 15 mm diameter (at widest point) stirrer is immersed to a depth of 7.5 cm and the 2 x 2 mm diameter titrant dosing tubes are immersed to a depth of 7.5 cm). The longer lag in natural seawater is discussed in section 3.3.4.

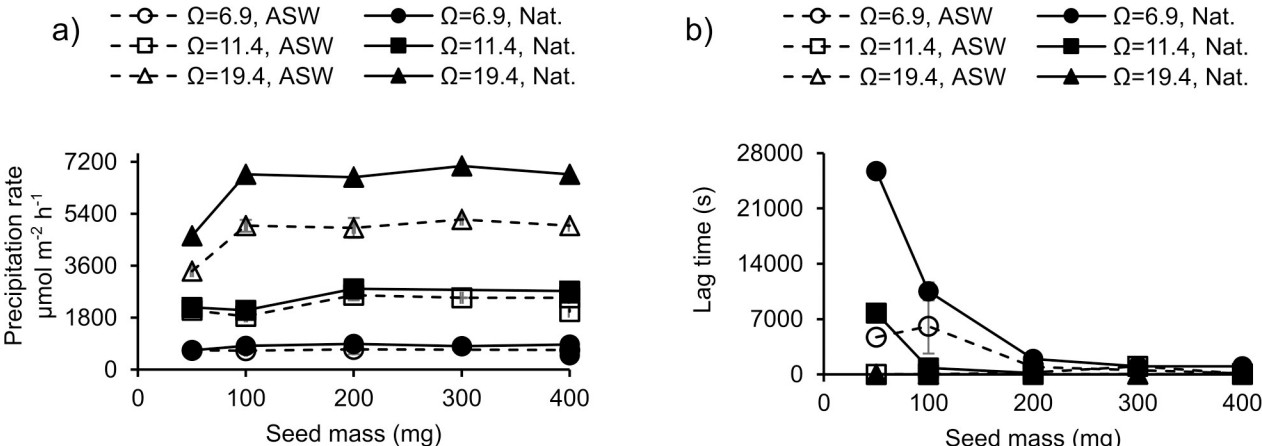

**Fig 6.** Estimated a) aragonite precipitation rate and b) lag period at start of precipitation in experiments comparing $\Omega$, starting seed mass and different waters. ASW = artificial seawater. Nat. = natural seawater. Error bars indicate 1 σ of duplicate precipitations. The reproducibility of estimated precipitation rates from duplicate experiments was typically 6% and was always <10%. (Reproducibility of lag periods was always better than 57%).

Precipitation rates normalised to seed surface area are comparable between duplicate experiments (typically 6%), and similar rates (i.e. within 6%) were estimated from experiments using 200, 300 and 400 mg seed. Precipitations normalised to 50 mg seed surface area yielded rates that were typically 70–80% of the rates observed at seed masses of 200–400 mg. Under only one set of conditions ($\Omega = 6.8$, artificial seawater) did precipitation rates from 50 mg agree with those from higher seed masses within duplicate error. It's not clear why precipitation rate is slower at small seed mass. We occasionally observed that some precipitate collected in a rim around the edge of the HDPE beaker during the experiment. This clumping may reflect an electrostatic attraction between the precipitate and the vessel and could reduce the $CaCO_3$ surface area in contact with the seawater and available for precipitation. Clumping is likely to have a larger effect at low seed mass when experiments are longer in duration (with more time to clump) and when any clumping will have a proportionally larger effect (due to the low starting seed mass).

**3.3.2 Effect of seed on Raman signature of *in vitro* precipitates.** We analysed the aragonite $v_1$ peak centre and FWHM of precipitates containing varying proportions of seed and *in vitro* precipitate (Fig 7). The peak centres and FWHM of all *in vitro* precipitates were significantly higher than that of the seed alone (ANOVA, $p \leq 0.05$). We observed no significant differences in the peak centre or FWHM between *in vitro* precipitates containing $\leq 40\%$ of the seed by mass. Our data suggests that the Raman signature of the *in vitro* precipitate can be resolved successfully if at least 60% of the analysed sample is precipitated *in vitro*.

**3.3.3 Optimising the aragonite precipitation method.** In this study we optimise a method for the precipitation of aragonite *in vitro* and normalise to the influences of seed mass and experiment duration on the final precipitate. Experiments with no seed are lengthy and any disequilibria between the atmospheric and seawater $CO_2$ can result in $CO_2$ invasion or outgas from the seawater solution creating a drift in both the [DIC] and $\Omega$ of the seawater solution. In this study unseeded experiments replicate poorly and exhibit different precipitation rates which may be an important control on aragonite geochemistry and structure. We consider unseeded experiments unsuitable for investigating the effects of environment and precipitation rate on geochemistry and structure. Adding a very small surface area/mass of seed can result in slow experiments which yield low precipitation rates (as observed in 50 mg seeded experiments above). Adding a larger surface area/mass of seed accelerates the experiment and we observed good agreement in precipitation rate estimates from experiments using 200–400 mg seed. Invasion and outgas of $CO_2$ is minor in these shorter experiments and measured DIC at the start and end of the precipitations using 200 mg seed agreed with predicted [DIC] within

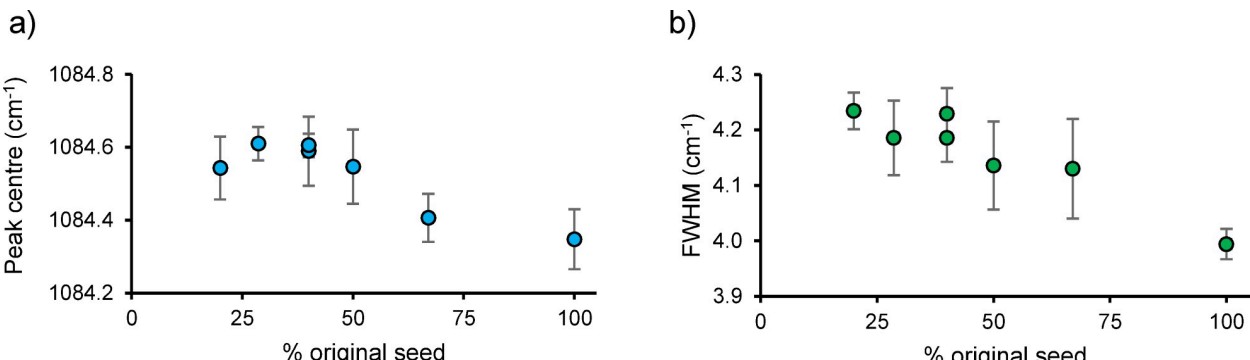

**Fig 7. Peak centre and FWHM of the v1 peak in the Raman spectra of precipitates containing different proportions of seed and *in vitro* precipitate.** a) Peak centre and b) Full width half maxima. The signatures of the seed are shown by the point at 100% seed. Points are means of 10 spectra and error bars are 1 standard deviation.

2% at $pH_{NBS}$ = 8.445 and 8.564, and within 3% at $pH_{NBS}$ = 8.337. However, increasing the amount of seed also increases the contamination of the geochemistry and structure of the final precipitate (seed plus *in vitro* precipitate) by the starting seed. Our analyses indicates that the Raman signature of the *in vitro* precipitate can be distinguished from the seed when the analysed sample is ≤40% of the seed by mass. In our experimental design this is equivalent to using a seed of 200 mg and precipitating 300 mg of aragonite *in vitro*. We have adopted this as our optimised methodology. Precipitating >300 mg *in vitro* will further reduce the domination of the precipitate by the seed, but it is likely that precipitating larger masses of aragonite has implications for the trace and minor element and isotopic compositions of the seawater solution used for the process of precipitation itself. It is all but impossible to produce titrant solutions that exactly replace the ions consumed during precipitation, particularly as element/ isotope partitioning is likely to vary under different experimental conditions (e.g. pH, Ω). For example, seawater Mg/Ca varied by 1–7% during precipitation of amorphous calcium carbonate from seawater under different pH, DIC chemistry and biomolecule availability [35]. The more aragonite precipitated from a solution, the larger the changes in the seawater composition will be, and this in itself will influence the partitioning of minor and trace elements and isotopes into aragonite.

**3.3.4 Effect of aspartic acid on aragonite precipitation rate and structure.** Aragonite precipitation rates were accelerated significantly by the addition of low concentrations of aspartic acid (by 1 and 10 μM in artificial and natural seawaters respectively) and inhibited by concentrations of ≥1 mM in both waters (ANOVA, $p \le 0.05$, Fig 8). Low concentrations of biomineralisation proteins [36], aspartic acid [37] and multiple residue aspartic acid peptides (aspartates) [38] have previously been found to promote the propagation of calcite crystals, while higher concentrations can inhibit propagation compared to controls [38]. Biomolecules play significant roles in controlling $CaCO_3$ precipitation and may operate by decreasing the energy barrier to ion attachment at the growing crystal face [38], or by blocking ion attachment [39], or by binding the dissolved $Ca^{2+}$ required for $CaCO_3$ precipitation [40]. Exquisite control over aragonite precipitation is required to produce the highly organised, regular skeletons deposited by corals [41]. Aspartic acid is the predominant amino acid in Scleractinia coral skeletons [17, 18] and is likely involved in the control of coral biomineralisation as the carboxyl acid side chain of this amino acid is negatively charged at physiological pH and may electrostatically attract $Ca^{2+}$ at the crystal surface [42]. Comparing the concentrations of aspartic acid incorporated in synthetic aragonite with those of coral skeletons suggests the [aspartic

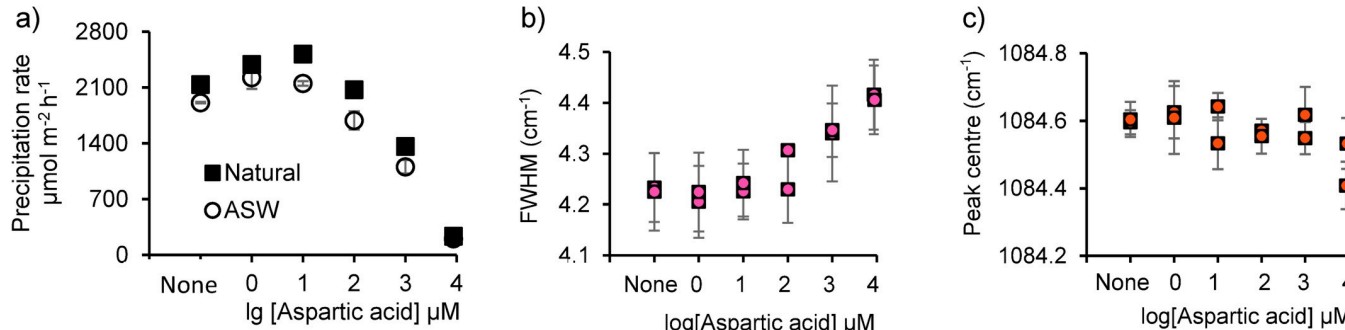

**Fig 8.** a) Aragonite precipitation rates from natural and artificial seawater in the presence of a range of concentrations of aspartic acid at Ω = 11.2. Precipitations were conducted in duplicate; points indicate mean rate and error bars indicate 1 standard deviation. The error bars are usually smaller than the symbols. b) FWHM and c) peak position of the $\nu_1$ peak in the Raman spectra of aragonites precipitated in the presence of aspartic acid. Points are means of 10–11 spectra and error bars are 1 standard deviation.

acid] of the coral calcification media is ~100–400 μM [18]. At these concentrations it is unclear if the amino acid acts to promote or inhibit aragonite growth (Fig 8). Aspartic acid predominantly occurs in coral skeletons as peptides and proteins, so further research is required to determine how these larger molecules influence aragonite precipitation. Low concentrations (0.1 μM) of large aspartic-rich peptides enhance calcite growth by a much greater magnitude than smaller peptides (of 6 aspartic acid residue or less) [38] but there is no current estimate of the likely concentrations of proteins at the coral calcification site to establish if these molecules truly promote or inhibit coral aragonite formation.

The concentrations of organic materials, [43, 44] aspartic acid in particular, [18] increase in the skeletons of corals cultured under high seawater $pCO_2$ (ocean acidification). If skeletal organics act to promote aragonite precipitation, then these increases could reflect a coral response to compensate for the reduced omega of seawater under ocean acidification, which typically inhibits coral calcification [44]. If skeletal organics inhibit aragonite precipitation, then this response to high seawater $pCO_2$ acts to intensify the reduction in coral calcification under ocean acidification [18].

Aragonite precipitation rates were significantly faster in natural compared to artificial seawater (paired t test, p = 0.0069), typically by ~19%. Although filtered before use, natural seawater contains a mixture of dissolved organic matter (DOM) not present in the artificial seawater which may influence $CaCO_3$ precipitation. Marine DOM is a complex mixture of biomolecules derived from marine and terrestrial sources (e.g. phytoplankton metabolism and plant decay products) and is typically 25–50% protein, 2–25% lipid and up to 40% carbohydrate [45]. Most seawater DOM has a molecular weight of <1 kDa (i.e. <1000 g per mole) as larger mass DOM is more readily biodegraded [46]. The majority of amino acids in the DOM occur as combined amino acids (i.e. as low molecular weight peptides, rather than free amino acids) [47]. Proteins smaller than 1 kDa contain ~10 amino acid residues or less and can be considered as peptides. Aspartic acid is one of the main constituents of the free and combined amino acids in seawater DOM [48] and can exceed concentrations of 1 μM [47]. In our study, [aspartic acid] of 1 μM promotes aragonite formation in artificial seawater and the likely inclusion of low levels of DOM in the natural seawater may explain why aragonite precipitation rates are typically higher in these waters. For this study the natural seawater was stored in the dark for several weeks before use. In the absence of phototropic organisms it is likely that the DOM will decrease as it is utilised by heterotrophic bacteria [47]. Although total dissolved amino acid likely decreased during storage, the contribution of seawater aspartic acid to the total amino acid likely increased, reflecting the discrimination against this amino acid during microbial heterotrophy [47]. We note that precipitation lag times (when they occur) are longer in natural compared to artificial seawater (Fig 6). High concentrations of aspartic acid (10 mM) can delay $CaCO_3$ nucleation and increase the solubility of initial $CaCO_3$ phases [49] but it is unclear if these effects are induced by DOM in seawater. The complexation of $Ca^{2+}_{(aq)}$ by DOM may also impact heterogenous nucleation.

Aragonites precipitated in the presence of aspartic acid were composed of larger, more pointer crystals than their counterparts precipitated in artificial seawater with no biomolecules (Fig 9). This suggests that the slower aragonite precipitation rates observed at higher concentrations of aspartic acid resulted in the formation of larger crystals.

Low concentrations of aspartic acid (1 and 10 μM) did not significantly affect the $v_1$ aragonite peak in the Raman spectra, but higher concentrations ($\geq$1 mM) caused an increase in FWHM of these bands compared to the aragonite precipitated with no aspartic acid (Fig 8, ANOVA, p<0.05). Aspartic acid additions did not affect the peak position. The $v_1$ band results from symmetric C-O stretching in the planar carbonate ion [50] and broadening of this peak is indicative of increased local disorder around the $CO_3^{2-}$ ion. We considered if broadening of

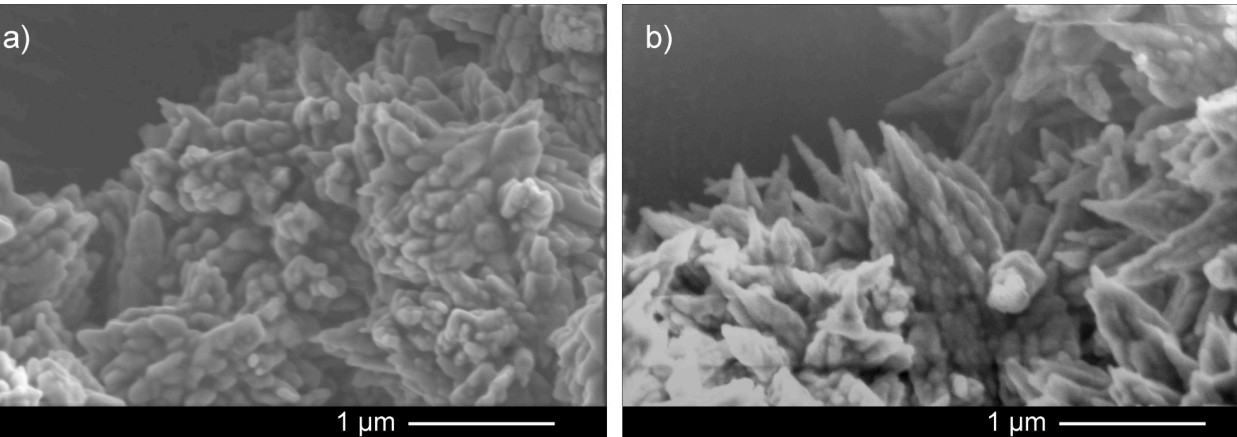

**Fig 9. Scanning electron micrographs of aragonite precipitated at $\Omega$ = 11.2.** a) without aspartic acid and b) with 8.7 mM aspartic acid. Scale bars are 1 μm.

the $\nu_1$ peak could reflect incorporation of another $CaCO_3$ phase e.g. calcite or amorphous calcium carbonate in the precipitated aragonite. Representative Raman spectra of all precipitates are included in the S1 Data. All spectra exhibit a pronounced dual peak at 700–710 cm$^{-1}$, indicative of aragonite [30]. Furthermore, the lattice mode vibrations observed at 100–250 cm$^{-1}$ in all precipitates are also consistent with aragonite with no evidence of the features associated with calcite or amorphous calcium carbonate [51]. Finally, we note that we have been unable to precipitate amorphous calcium carbonate in our laboratory at the pH and $\Omega$ tested here, even in the presence of aspartic acid [52]. Collectively, we find no evidence that non-aragonitic CaCO3 phases contribute to $\nu_1$ peak broadening. The primary cause of the loss of short-range order in the precipitates remains unclear. Rotational disorder can be caused by rapid disequilibrium crystal growth [4] or can be a consequence of the incorporation of contaminant ions in the crystal lattice which create local lattice distortions [3].

Whatever its cause, the Raman aragonite $\nu_1$ peak FWHM has been linked to increases in the $\Omega$ of the seawater solution used for precipitation [4] and to the inclusion of biomolecules [53]. Similarly, in coral skeletons, broadening of the v1 peak is observed in the centres of calcification (the features at the centres of the skeletal units) and may reflect increased calcification media omega [54] or increased skeletal organic material [55]. The FWHM of the aragonite v1 peak has previously been used to infer the saturation state of the coral calcification fluid [4] and to track the calcification response of corals to increased seawater $pCO_2$. However, our study demonstrates that the inclusion of biomolecules in aragonite can influence aragonite disorder even at constant fluid saturation state. Therefore, the increases in the organic and amino acid contents of skeletons of corals cultured at high seawater $pCO_2$ [18, 43, 44] are likely to influence the structural disorder of the aragonite skeletons and potentially obscure any calcification fluid/seawater $\Omega$ signal. Further work is required to clarify how the inclusion of aspartic acid in peptides and proteins (as in biogenic aragonites) affects the aragonite precipitation rate and structure.

## 4. Conclusions

We optimised a method for the precipitation of synthetic aragonites under simulated biological conditions by using aragonite seed as a substrate to improve reproducibility of precipitation rate in comparison with unseeded experiments. Initial application of this method shows that

aspartic acid, the most common amino acid in Scleratinian coral skeletons, promotes aragonite precipitation at low concentrations (1 and 10 μM) but inhibits precipitation at concentrations ≥1mM. Aragonite crystals precipitated in the presence of high concentrations of aspartic acid have wider FWHM of the Raman spectrum $v_1$ peak (indicative of the carbonate ion symmetric stretch) suggesting that the biomolecule disrupts the rotational structural order of the aragonite lattice. Future changes in seawater chemistry (influenced by climate) may alter the current ratios of organic molecules in seawater and therefore alter the contributions of aspartic acid (and other amino acids) to coral skeletons, highlighting the relevance and importance of being able to simulate precipitation conditions and scenarios. A standard and reliable precipitation method is necessary to further investigate the influence of changing environmental conditions on the geochemistry of coral skeletons, and to produce comparable results and advance research in this field.

## Supporting information

**S1 Fig. Raman of aspartic acid.**
(DOCX)

**S1 Data. This file consists of the supporting data tables.**
(XLSX)

## Acknowledgments

We thank David Miller, Aaron Naden and Gavin Peters at the University of St Andrews, for their assistance with the Raman, SEM and BET analyses respectively.

## Author Contributions

**Conceptualization:** Adrian Finch, Kirsty Penkman, Roland Kröger, Nicola Allison.

**Data curation:** Celeste Kellock, Maria Cristina Castillo Alvarez, Nicola Allison.

**Formal analysis:** Celeste Kellock, Maria Cristina Castillo Alvarez, Nicola Allison.

**Funding acquisition:** Adrian Finch, Kirsty Penkman, Roland Kröger, Matthieu Clog, Nicola Allison.

**Investigation:** Celeste Kellock, Maria Cristina Castillo Alvarez, Nicola Allison.

**Methodology:** Celeste Kellock, Maria Cristina Castillo Alvarez, Nicola Allison.

**Project administration:** Nicola Allison.

**Supervision:** Nicola Allison.

**Writing – original draft:** Celeste Kellock, Nicola Allison.

**Writing – review & editing:** Celeste Kellock, Maria Cristina Castillo Alvarez, Adrian Finch, Kirsty Penkman, Roland Kröger, Matthieu Clog, Nicola Allison.

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
