## [Decision Letter · Decision Letter 0]

8 Sep 2022

PONE-D-22-18432Optimising a method for aragonite precipitation in simulated biogenic calcification mediaPLOS ONE

Dear Dr. Kellock,

Thank you for submitting your manuscript to PLOS ONE. After careful consideration, we feel that it has merit but does not fully meet PLOS ONE’s publication criteria as it currently stands. Therefore, we invite you to submit a revised version of the manuscript that addresses the points raised during the review process.

We look forward to receiving your revised manuscript.

Kind regards,

Jinhui Tao, Ph.D.

Academic Editor

PLOS ONE

Journal Requirements:

2. Please expand the acronym “EPSRC” (as indicated in your financial disclosure) so that it states the name of your funders in full.

   "This work was supported by the Leverhulme Trust (Research project grant 2015-268 to NA, RK, and KP) and the UK Natural Environment Research Council (NE/S001417/1) to NA, KP, RK, MC and AF). We thank David Miller and Gavin Peters, both at the University of St Andrews, for their assistance with the Raman and BET analyses respectively. The Raman Microscope is supported by the EPSRC (Light Element Analysis Facility Grant EP/T019298/1 and Strategic Equipment Resource Grant EP/R023751/1)."

 "This work was supported by the Leverhulme Trust (Research project grant 2015-268 to NA, RK, and KP) and the UK Natural Environment Research Council (NE/S001417/1) to NA, KP, RK, MC and AF).

The Raman Microscope is supported by the EPSRC (Light Element Analysis Facility Grant EP/T019298/1 and Strategic Equipment Resource Grant EP/R023751/1). 

Additional Editor Comments:

All the reviewers recognize the merit of the manuscript and they recommend that it needs to be carefully revised before the acceptance. I suggest the author read the reviewer's comments (in the attachment) and revise it point-by-point accordingly. The highlight in the revised part are highly preferred for our review.

Reviewers' comments:

Reviewer's Responses to Questions

**Comments to the Author**

1. Is the manuscript technically sound, and do the data support the conclusions?

Reviewer #1: No

Reviewer #2: Yes

Reviewer #3: Partly

2. Has the statistical analysis been performed appropriately and rigorously? 

Reviewer #1: N/A

Reviewer #2: Yes

Reviewer #3: Yes

3. Have the authors made all data underlying the findings in their manuscript fully available?

Reviewer #1: Yes

Reviewer #2: Yes

Reviewer #3: Yes

4. Is the manuscript presented in an intelligible fashion and written in standard English?

Reviewer #1: No

Reviewer #2: Yes

Reviewer #3: Yes

5. Review Comments to the Author

Reviewer #1: I think the goal of the paper is important and interesting to a specific group working on CaCO3 mineralization. However, I find it quite difficult to follow the paper. I think the paper can be restructured and wrote more concisely and logically. Also, some figures are too sloppy. For example,

1. The x axis in the Figure 3b seems wrong. Why two axes are showed here?

2. Figure 6 does not have alphabets. I guess I can figure it out myself, but that’s annoying. What’s the unit of precipitation rate here? The same as the one mentioned before? ( btw, I was very interested in the lag time measurement, but find it difficult to follow in the paper, also, when you reference to the figure, please be more precise. In many cases, you only refer to figure xx. It is better to go to a specific panel.

3. Figure 8a. obvious typos.

4. Figure 9, don’t use “left” and “right” for the figures.

Overall, I think the paper needs to be carefully/majorly revised.

Reviewer #2: In this study, the authors synthesized aragonite in simulated biogenic calcification solution using a titration system and investigated the influence of aspartic acid on the structure and morphology. They found that the seeded crystallization experiment was more reproducible than unseeded experiments and that the full width half maxima of the v1 peak in Raman increases with increasing aspartic acid concentration. Firstly, I do not gain much new knowledge regarding the preparation method of aragonite, as claimed by the title of the manuscript, although they show that by adding seeds in the solution the experiments are more reproducible. I suggest the authors to change the title to better reflect the content of the work, which should be focusing on the precipitation rate and the influence of aspartic acid. Secondly, it is well known that the incorporation of aspartic acid in calcite could results a change of the lattice parameter, and may change the FWHM of the Raman peaks. It is kind of interesting to see such a change in aragonite, suggesting either the crystallinity of aragonite decreases or aspartic acid is incorporated in the structure of aragonite. The authors should perform more chemical and structural characterization, such amino acid analysis, XRD, infrared, and provide some explanations. Therefore, I do not recommend publication of this manuscript in its current form.

Reviewer #3: The authors present a well written article with a systematic experimental approach involving the use of Raman spectroscopy. A reliable method to study the impact of various organic and inorganic components would be highly beneficial to better understand geochemistry and calcified marine life through various lifecycles. However, this article needs major revision or resubmission before further consideration for the following reasons.

1. The references do not match the statements in main text and there seems to be an extra reference somewhere. This makes it difficult to read the manuscript and understand what the manuscript is talking about or verify the validity of the statements. For example:

Line 87-91: Ref 16 does not use Raman spectroscopy nor discusses CaCO3, though the statement could be talking about reference 15.

Line 418 – Ref 35 uses no seawater in their study

Line 427 to 433, references don’t match the statement.

2. Line 87-91: Reference 3 discusses rotational disorder from Mg or other impurities cannot be easily detected from aragonite but can be detected from Mg-calcite, which argues against the authors use of Raman for detecting disorder as a result of Sr or organics in aragonite using raman.

3. Figure 2. I’m concerned about the use of the v1 peak as an identifier for the presence of aragonite because this peak seems to exists for all polymorphs of CaCO3. It could be useful to see a combined full spectrum for all the samples/measurements against a reference aragonite, calcite, ACC spectra etc , somewhere in the SI or repository, especially at 100 -350 cm-1- wavenumber which can better identify the polymorph than the v1 peak (Ref: https://www.nature.com/articles/s41467-018-07601-3 ). Because it is hard to believe that precipitates other than aragonite are not produced at such high supersaturations or by the ex situ method of analysis involving drying of the precipitates. Is it possible that presence of precipitates other than aragonite could cause small changes in that peak? Especially a broadening caused by the presence of amorphous calcium carbonate which have relatively more disorder around CO3 compared to aragonite, rather than the presence of polyaspartic acid as the authors suggests later on. ACC formation is a common intermediate when using highly charged polymers which form polymer-induced liquid precursor (PILP) droplets in presence of cations and anions. In the PILP method, the polymer is typically excluded from the crystal that forms from aggregates of ACC+polymer nanoparticles (Ref: https://www.nature.com/articles/s41467-018-05006-w ); this exclusion of the polymer also conflicts with the author’s interpretation that occlusion of aspartic acid in the aragonite causes peak broadening

4. Line 246 to 262. It is understandable that the precipitation is not reproducible without the presence of aragonite seeds, however, this section is confusing to me, especially relationship between the supersaturation, [DIC] and CO2. Why does the higher omega sample produce lower precipitation? One would expect it to be the other way around.

5. Line 483 – 503: Following point#3 above, is there any evidence to show that higher amorphous mineral is not formed with higher aspartic acid concentrations and results in the peak broadening? Aspartic acid is too large to cause lattice level disorder or incorporation into the lattice that was previously reported for Mg incorporation into calcite lattice.

6. Line 521: It would be helpful for the reader to have the conclusion clearly state how the article has optimized the precipitation. For example, “We optimized a method for the precipitation of synthetic aragonites under simulated biological conditions by using aragonite seeds as substrate to improve reproducibility of precipitation rates” or better

6. PLOS authors have the option to publish the peer review history of their article (what does this mean?). If published, this will include your full peer review and any attached files.

Reviewer #1: No

Reviewer #2: No

Reviewer #3: No

---

## [Author Response · Author response to Decision Letter 0]

22 Sep 2022

PONE-D-22-18432

Response to academic editor and reviewers.

We’ve pasted the comments by the academic editor and have responded below. 

Academic editor:

Manuscript meets PLOS ONE’s style requirements.

Done.

2. Please expand the acronym “EPSRC” (as indicated in your financial disclosure) so that it states the name of your funders in full. This information should be included in your cover letter; we will change the online submission form on your behalf.

Please find the acronym expanded in the cover letter as requested.

3. Thank you for stating the following in the Acknowledgments Section of your manuscript.

We have removed the funding information from the manuscript. Funding information should read: This research has been supported by the Leverhulme Trust (Research project grant 2015-268 to NA, RK, and KP) and the UK Natural Environment Research Council (NE/S001417/1 to NA, KP, RK, MC and AF). The Raman Microscope is supported by the Engineering and Physical Sciences Research Council (Light Element Analysis Facility Grant EP/T019298/1 and Strategic Equipment Resource Grant EP/R023751/1#).

Captions for the supplementary data, and supporting data, files / figures / tables have been listed at the end of the manuscript as requested. Guidance states ‘all file types can be submitted, but files must be smaller than 20MB in size’, we have adhered to the file size limit and are submitting one word document and one excel booklet. 

We have pasted the reviewers comments and responded below (-).

Reviewer #1: 

I think the goal of the paper is important and interesting to a specific group working on CaCO3 mineralization. However, I find it quite difficult to follow the paper. I think the paper can be restructured and wrote more concisely and logically. Also, some figures are too sloppy. For example,

1. The x axis in the Figure 3b seems wrong. Why two axes are showed here?

-We have revised this figure. Figure 3b shows a high-resolution image of Figure 3a and two horizontal axes are shown, one for each titrator. We have labelled each axis to make this clear.

To make the text clearer, there have been some structural changes / inserts in the text: (Lines 59, 94, 514, 525, 559, 570 in the Tracked Changes document) (Lines 59, 94, 489, 500, 530, 541 in the Revised Manuscript document)

2. Figure 6 does not have alphabets. I guess I can figure it out myself, but that’s annoying. What’s the unit of precipitation rate here? The same as the one mentioned before? ( btw, I was very interested in the lag time measurement, but find it difficult to follow in the paper, also, when you reference to the figure, please be more precise. In many cases, you only refer to figure xx. It is better to go to a specific panel.

-We have revised this figure, to label the units on the precipitation axis and label the sub panels. 

Please find panel specific references in the text (Line 244, 247, 348 on the tracked changes document). (Lines 238, 241, 336 on the Revised manuscript with accepted changes document).

3. Figure 8a. obvious typos.

-We have labelled the sub panels in this figure and adjusted the figure caption accordingly.

4. Figure 9, don’t use “left” and “right” for the figures.

-We have labelled the sub panels in this figure and adjusted the figure caption accordingly.

Overall, I think the paper needs to be carefully/majorly revised.

-We have labelled all the figure sub panels and have revised the main text to refer to figures by sub panel where appropriate to make the meaning of the text clearer. 

Reviewer #2:

In this study, the authors synthesized aragonite in simulated biogenic calcification solution using a titration system and investigated the influence of aspartic acid on the structure and morphology. They found that the seeded crystallization experiment was more reproducible than unseeded experiments and that the full width half maxima of the v1 peak in Raman increases with increasing aspartic acid concentration. Firstly, I do not gain much new knowledge regarding the preparation method of aragonite, as claimed by the title of the manuscript, although they show that by adding seeds in the solution the experiments are more reproducible. I suggest the authors to change the title to better reflect the content of the work, which should be focusing on the precipitation rate and the influence of aspartic acid. 

-In this paper we report research which focuses on developing a reproducible method for precipitating aragonite under simulated biological conditions. The manuscript title reflects this. 

Secondly, it is well known that the incorporation of aspartic acid in calcite could results a change of the lattice parameter, and may change the FWHM of the Raman peaks. It is kind of interesting to see such a change in aragonite, suggesting either the crystallinity of aragonite decreases or aspartic acid is incorporated in the structure of aragonite. The authors should perform more chemical and structural characterization, such amino acid analysis, XRD, infrared, and provide some explanations. Therefore, I do not recommend publication of this manuscript in its current form.

-This is primarily a methods development paper, in which optimise a method for the precipitation of aragonite in seawater under conditions which are similar to those of biogenic calcification media. We demonstrate the application of the method by precipitating aragonite in the presence of aspartic acid. We show that aspartic acid can both promote and inhibit aragonite precipitation (depending on concentration) and that high aspartic acid concentrations increase the rotational disorder in the aragonite structure. We have not undertaken other characterisation of the effects of aspartic acid. 

Reviewer #3: 

The authors present a well written article with a systematic experimental approach involving the use of Raman spectroscopy. A reliable method to study the impact of various organic and inorganic components would be highly beneficial to better understand geochemistry and calcified marine life through various lifecycles. However, this article needs major revision or resubmission before further consideration for the following reasons.

1. The references do not match the statements in main text and there seems to be an extra reference somewhere. This makes it difficult to read the manuscript and understand what the manuscript is talking about or verify the validity of the statements. For example:

Line 87-91: Ref 16 does not use Raman spectroscopy nor discusses CaCO3, though the statement could be talking about reference 15.

-We mis-numbered the references. We have checked this and ensured the reference numbering is now correct. 

Line 418 – Ref 35 uses no seawater in their study. This statement applies to reference 34 in the submitted manuscript. 

-We have revised the reference numbering to correct this. 

Line 427 to 433, references don’t match the statement.

-We have revised the reference numbering to correct this.

2. Line 87-91: Reference 3 discusses rotational disorder from Mg or other impurities cannot be easily detected from aragonite but can be detected from Mg-calcite, which argues against the authors use of Raman for detecting disorder as a result of Sr or organics in aragonite using Raman.

-This statement in the introduction refer to both references 3 and 4 (line 90-94 in the Tracked Changes document and the Revised Manuscript document). Reference 3 (Kameneos et al., 2013) discusses high Mg calcite. Reference 4 (De Carlo et al., 2017) reports aragonite and identifies a correlation between the FWHM of the aragonite Raman ν1 peak and the mean Ω of the precipitating fluid and concludes that this reflects increased carbonate rotational disorder.

3. Figure 2. I’m concerned about the use of the v1 peak as an identifier for the presence of aragonite because this peak seems to exists for all polymorphs of CaCO3. It could be useful to see a combined full spectrum for all the samples/measurements against a reference aragonite, calcite, ACC spectra etc , somewhere in the SI or repository, especially at 100 -350 cm-1- wavenumber which can better identify the polymorph than the v1 peak (Ref: https://www.nature.com/articles/s41467-018-07601-3 ). Because it is hard to believe that precipitates other than aragonite are not produced at such high supersaturations or by the ex situ method of analysis involving drying of the precipitates. Is it possible that presence of precipitates other than aragonite could cause small changes in that peak? Especially a broadening caused by the presence of amorphous calcium carbonate which have relatively more disorder around CO3 compared to aragonite, rather than the presence of polyaspartic acid as the authors suggests later on. ACC formation is a common intermediate when using highly charged polymers which form polymer-induced liquid precursor (PILP) droplets in presence of cations and anions. In the PILP method, the polymer is typically excluded from the crystal that forms from aggregates of ACC+polymer nanoparticles (Ref: https://www.nature.com/articles/s41467-018-05006-w ); this exclusion of the polymer also conflicts with the author’s interpretation that occlusion of aspartic acid in the aragonite causes peak broadening

-We identify the precipitates as aragonite using the Raman spectrum ν4 peak (line 230 in both the Tracked Changes and Revised Manuscript documents, Figure 2), not the v1 peak. We have added a section to the supplementary data to show representative Raman spectra for each of the precipitates formed in the presence and absence of aspartic acid. We discuss this data and the possibility that non-aragonite CaCO3 phases contribute to v1 peak broadening (Lines 514-524 in the Tracked Changes document and lines 489-498 in the Revised Manuscript document) i.e. 

‘We considered if broadening of the ν1 peak could reflect incorporation of another CaCO3 phase e.g. calcite or amorphous calcium carbonate in the precipitated aragonite. Representative Raman spectra of all precipitates are included in the Supplementary data. All spectra exhibit a pronounced dual peak at 700-710 cm-1, indicative of aragonite30. Furthermore, the lattice mode vibrations observed at 100-250 cm-1 in all precipitates are also consistent with aragonite with no evidence of the features associated with calcite or amorphous calcium carbonate51.Finally we note that we have been unable to precipitate amorphous calcium carbonate in our laboratory at the pH and Ω tested here, even in the presence of aspartic acid52. Collectively, we find no evidence that non-aragonitic CaCO3 phases contribute to ν1 peak broadening. The primary cause of the loss of short-range order in the precipitates remains unclear. Rotational disorder can be caused by rapid disequilibrium crystal growth4 or can be a consequence of the incorporation of contaminant ions in the crystal lattice which create local lattice distortions3 .’

4. Line 246 to 262. It is understandable that the precipitation is not reproducible without the presence of aragonite seeds, however, this section is confusing to me, especially relationship between the supersaturation, [DIC] and CO2. Why does the higher omega sample produce lower precipitation? One would expect it to be the other way around.

-The unseeded experiments conducted at higher Ω (Figure 3, line 264 Tracked Changes document and line 258 Revised Manuscript document) result in faster aragonite precipitation than at low Ω. Contrast the precipitation at high and low Ω in Figure 3. Precipitations at high Ω occur faster than those at low Ω. We do not calculate precipitation rates as we do not know the surface area of the surface on which aragonite precipitation eventually proceeds. 

5. Line 483 – 503: Following point#3 above, is there any evidence to show that higher amorphous mineral is not formed with higher aspartic acid concentrations and results in the peak broadening? Aspartic acid is too large to cause lattice level disorder or incorporation into the lattice that was previously reported for Mg incorporation into calcite lattice.

-We address this point in response to 3. above. Specifically, find no evidence of amorphous calcium carbonate in the Raman spectra and we have been unable to produce amorphous calcium carbonate under the Ω used in these experiments, even in the presence of aspartic acid (Lines 518 – 524 in Tracked changes document, and lines 493 – 498 in the Revised Manuscript document). The cause of the increased carbonate disorder at higher aspartic acid concentrations is currently unknown (Lines 524 – 525 in the Tracked Changes document and line 499 in the Revised Manuscript document).

6. Line 521: It would be helpful for the reader to have the conclusion clearly state how the article has optimized the precipitation. For example, “We optimized a method for the precipitation of synthetic aragonites under simulated biological conditions by using aragonite seeds as substrate to improve reproducibility of precipitation rates” or better

-We have revised the conclusions to include the above statement (Lines 599 – 560 in the Tracked Changes document and lines 530 – 531 in the Revised Manuscript document). We have also added a final sentence to the end of the conclusion to explain how this method will advance the field i.e. ‘A standard and reliable precipitation method is necessary to further investigate the influence of changing environmental conditions on the geochemistry of coral skeletons, and to produce comparable results and advance research in this field’ (lines 570 – 572 in the Tracked Changes document and lines 541 – 543 in the Revised Manuscript document).

---

## [Decision Letter · Decision Letter 1]

21 Nov 2022

Optimising a method for aragonite precipitation in simulated biogenic calcification media

PONE-D-22-18432R1

Dear Dr. Kellock,

We’re pleased to inform you that your manuscript has been judged scientifically suitable for publication and will be formally accepted for publication once it meets all outstanding technical requirements.

Kind regards,

Amitava Mukherjee, ME, Ph.D.

Academic Editor

PLOS ONE

Additional Editor Comments (optional):

Reviewers' comments:

Reviewer's Responses to Questions

**Comments to the Author**

1. If the authors have adequately addressed your comments raised in a previous round of review and you feel that this manuscript is now acceptable for publication, you may indicate that here to bypass the “Comments to the Author” section, enter your conflict of interest statement in the “Confidential to Editor” section, and submit your "Accept" recommendation.

Reviewer #1: All comments have been addressed

Reviewer #2: All comments have been addressed

Reviewer #3: All comments have been addressed

2. Is the manuscript technically sound, and do the data support the conclusions?

Reviewer #1: Yes

Reviewer #2: Yes

Reviewer #3: Yes

3. Has the statistical analysis been performed appropriately and rigorously? 

Reviewer #1: N/A

Reviewer #2: Yes

Reviewer #3: Yes

4. Have the authors made all data underlying the findings in their manuscript fully available?

Reviewer #1: Yes

Reviewer #2: Yes

Reviewer #3: Yes

5. Is the manuscript presented in an intelligible fashion and written in standard English?

Reviewer #1: Yes

Reviewer #2: Yes

Reviewer #3: Yes

6. Review Comments to the Author

Reviewer #1: (No Response)

Reviewer #2: The authors have addressed all my concerns and I have no further comments. I suggest publication of the manuscript.

Reviewer #3: (No Response)

7. PLOS authors have the option to publish the peer review history of their article (what does this mean?). If published, this will include your full peer review and any attached files.

Reviewer #1: No

Reviewer #2: No

Reviewer #3: No

---

## [Editor Report · Acceptance letter]

25 Nov 2022

PONE-D-22-18432R1 

Optimising a method for aragonite precipitation in simulated biogenic calcification media 

Dear Dr. Kellock:

I'm pleased to inform you that your manuscript has been deemed suitable for publication in PLOS ONE. Congratulations! Your manuscript is now with our production department. 

Kind regards, 

on behalf of

Professor Dr. Amitava Mukherjee 

Academic Editor

PLOS ONE